

# A comprehensive exploration of human communal media interaction and its evolving impact on psychological health across demographics and time

Tajim Md. Niamat Ullah Akhund[1,2], Deep Ajabani[3], Zaffar Ahmed Shaikh[4,5], Ali Elrashidi[6], Waleed A. Nureldeen[7], Muhammad Ishaq Bhatti[8] and M Mesbahuddin Sarker[9]

[1] Department of CSE, Daffodil International University, Dhaka, Bangladesh
[2] Graduate School of Science and Engineering, Saga University, Saga, Japan
[3] Source InfoTech Inc., Loganville, GA, United States of America
[4] Department of Computer Science and Information Technology, Benazir Bhutto Shaheed University Lyari, Karachi, Sindh, Pakistan
[5] School of Engineering, École Polytechnique Federale de Lausanne, Lausanne, Switzerland
[6] Electrical Engineering Department, University of Business and Technology, Jeddah, Saudi Arabia
[7] UBT Academy, University of Business and Technology, Jeddah, Saudi Arabia
[8] Department of Economics, Finance and Marketing, La Trobe University, Melbourne, NSW, Australia
[9] Institute of Information Technology, Jahangirnagar University, Dhaka, Bangladesh

Corresponding authors
Zaffar Ahmed Shaikh,
zashaikh@bbsul.edu.pk
Ali Elrashidi, a.elrashidi@ubt.edu.sa

## ABSTRACT

This work presents a comprehensive literature review and real-world survey analysis examining the intricate relationship between communal media and psychological concerns within human–computer interaction (HCI). By systematically categorizing research from 1995 to 2023, we observe a significant increase in studies, reflecting the growing recognition of communal media's impact on psychological well-being. Our review delves into various HCI aspects, including demographic effects, emotional sentiment analysis, online social networking, and communal media use intensity. A real-world survey focusing on TikTok users reveals diverse user perspectives on safety, privacy, and the platform's impact on creativity and mental health. The data highlights a demographic predominance of young adults aged 16–24, with creativity, leisure, and content discovery being primary engagement motivators. Positive impacts such as creative inspiration and distraction from worries are contrasted with concerns over privacy invasion, harm to underaged users, and addiction. Correlation analysis under-scores the complex interplay between user experiences and perceptions. Future research should prioritize enhancing privacy and safety measures, educational campaigns, and in-depth studies on the long-term psychological impacts of TikTok usage. Additionally, exploring platform modifications and conducting longitudinal and cross-platform studies will help create a safer and more engaging social media environment. This work contributes significantly to the expanding HCI body of knowledge, providing a robust foundation for future research to inform policies and interventions promoting positive psychological outcomes in the digital age.

# INTRODUCTION

The rapid rise of communal media platforms has transformed how we communicate and express ourselves online. These platforms, designed to enhance user experiences, are now essential tools for navigating the complexities of the digital age. However, concerns about their impact on our psychological well-being persist. Researchers, particularly focusing on young adults, stress the need to examine the psychological implications of communal media engagement (*Primack et al., 2017*). Increased use of communal media has been linked to social isolation, highlighting the importance of understanding the consequences of digital connectivity. As individuals navigate these digital spaces, they can be affected by various factors. *Kross et al. (2013)* found that prediction algorithms within communal media platforms can have harmful effects on users. This challenges the assumption that more connectivity always leads to better well-being. In this review, we explore the effects of communal media across different demographics, such as adolescents, adults, and older adults (*Moreno et al., 2011*). For example, *Moreno et al. (2011)* investigates how communal media interfaces impact depression disclosures among college students, stressing the need to understand these effects on specific groups. The review also analyzes online validation seeking (*Naslund et al., 2016*), contributing to our understanding of the complex relationship between digital engagement and mental well-being. In this work, we consider communal media as all types of online social networking sites and communication mediums. This work also uncovers the dynamics of communal media through a real-world survey. The survey showed a varied user landscape with distinct opinions about safety, privacy, and the influence of the chosen communal media platform on creativity and mental health.

In today's world, communal media has become a huge part of our daily lives, changing the way we connect and share information (*Braghieri, Levy & Makarin, 2022*). This shift in how we communicate has led to a significant transformation in our society. Researchers have been carefully examining the various ways communal media influences our mental well-being, touching on aspects such as our thoughts, social interactions, and emotional experiences. The integration of communal media has been so widespread that it has brought about a revolutionary era in communication (*Naslund et al., 2020*). Imagine communal media as a giant web woven into our daily activities, connecting people from all walks of life (*Pantic, 2014*). This change in the way we communicate has prompted researchers to dive deep into understanding the intricate impact of communal media on our psychological concerns. When we talk about psychological concerns, we are referring to how our minds work, how we feel, and how we relate to others (*Bashir & Bhat, 2017*). Researchers have not just focused on one aspect but have explored a range of dimensions, including psychological, social, and emotional aspects (*Sadagheyani & Tatari, 2020*; *Karim et al., 2020*). It's like

peeling layers of an onion to reveal different facets of how communal media affects us. Numerous studies, such as *Alfiah et al. (2021)*, *Zsila & Reyes (2023)* and *Valkenburg, Meier & Beyens (2022)* have contributed to this growing body of knowledge. These studies delve into the impact of communal media on our minds, uncovering the complex ways it influences our thoughts and emotions. For instance, researchers have looked at the effects of online interactions, exploring how they shape our social lives and impact our overall well-being. The exploration doesn't stop there; it extends to understanding the psychological, social, and emotional dimensions (*Meier & Reinecke, 2021*). *Barthorpe et al. (2020)* and *OKeeffe & Clarke-Pearson (2011)* have added valuable insights, examining the effects of communal media engagement on our psychological state. It's like studying a complex puzzle where each piece represents a different aspect of our mental well-being. communal media has become a vast landscape with platforms like Facebook, Twitter, and Instagram, each contributing to this transformative era (*Jones, Mougouei & Evans, 2021*; *Kelly et al., 2018*). *Fiori, Antonucci & Cortina (2006)*, *Conway & OConnor (2016)* and *Nicholas, Onie & Larsen (2020)* have explored the pros and cons, highlighting the diverse impacts of communal media on our minds. It's akin to navigating through a digital terrain where the terrain itself shapes our thoughts and feelings. As we journey through the research, we encounter *Bekalu, McCloud & Viswanath (2019)* and *Kawachi & Berkman (2001)*, each providing a piece of the puzzle. These studies shed light on the social dynamics of online platforms and their role in shaping our psychological landscape (*Deepa & Priya, 2020*). It's like understanding the dynamics of a social ecosystem where every interaction leaves an imprint on our minds. *Conway & OConnor (2016)* and *Nicholas, Onie & Larsen (2020)* contribute to this exploration, emphasizing the need to understand the ethical dimensions of communal media research. It's not just about studying the impact; it's about doing so responsibly, considering the potential effects of the research itself on individuals. *Kawachi & Berkman (2001)* showed a closer look at the association between communal media use and psychological concerns. It's like unraveling a mystery, understanding how our engagement with communal media links to our mental well-being. These studies highlight the intricate connections between our online lives and our psychological states. In this vast landscape of research, we also encounter studies by *Kawachi & Berkman (2001)*, which explores the broader social implications of our digital interactions (*Barthorpe et al., 2020*). It's like zooming out to see the bigger picture, understanding how our individual experiences on communal media contribute to the overall social fabric. As we navigate this sea of information, it becomes clear that understanding the impact of communal media on our psychological concerns is a multifaceted endeavor (*Scott & Woods, 2019*). It's not a simple equation but a complex interplay of factors that shape our thoughts, feelings, and social connections (*Bekalu, McCloud & Viswanath, 2019*). Works addressing the effects of global events, such as the impact of COVID-19 on education and the development of Internet of Things-based support systems (*Akhund, 2023*; *Akhund, Newaz & Sarker, 2024*), further highlight the pivotal role played by the Internet of Things in addressing real-world challenges (*Akhund et al., 2024*). In the realm of human–computer interaction, the integration of Internet of Things technologies has given rise to low-cost solutions for health screening, posture control, and robotic assistance (*Akhund et al., 2022b*; *Akhund*

*et al., 2022a*; *Akhund et al., 2021*). Additionally, studies examining the role of robotics in medical applications and smart agricultural systems using the Internet of Things emphasize the interdisciplinary nature of these technological domains (*Himel et al., 2022*; *Suny et al., 2022*). Security perspectives in the Internet of Things (*Newaz et al., 2021*), parallelization of image processing algorithms (*Mia et al., 2022*), and the development of self-powered Internet of Things-based solutions for poultry farming (*Akhund et al., 2020*) collectively contribute to shaping a comprehensive and secure human–computer interaction landscape. The journey through these studies is like embarking on an exploration of the mind, with each research paper adding a layer of understanding to the complex relationship between communal media and our mental well-being.

## Contributions

In light of these considerations, the literature review provides a comprehensive synthesis of existing research, offering insights into human communal media interaction and its effects on psychological concerns. The review also informs future research directions and contributes to evidence-based strategies promoting positive psychological concerns in the digital age. Our additional contributions encompass a systematic categorization of research works, statistical and mathematical analyses, and future-oriented recommendations, enhancing the depth and breadth of this comprehensive exploration. The contributions of this work are as follows:

1. Analyzes negative phenomena related to communal media, including cyber-bullying, social comparison, and online validation seeking, crucial for devising targeted interventions from the aspects of human–computer interaction.
2. Explores the impact across demographic groups, considering factors such as gender, culture, and age with the investigation of emerging trends.
3. Identifies gaps in current research and suggests potential directions for future investigations, encouraging nuanced exploration of the communal media and psychological concerns nexus.
4. Critically evaluates interventions, contributing to the discourse on best practices and ethical considerations in utilizing digital platforms for psychological concerns promotion and support.
5. This study sheds light on social media community dynamics through a real-world survey. It aims to unveil the intricacies of these dynamics for a comprehensive understanding.

## Overview of the paper

This paper begins with an introduction section. After that, a detailed methodology section is illustrated that outlines the approach used for both systematic literature review and real-world survey methodologies. The ''Results and Discussion of Literature Survey'' section presents findings from the systematic review conducted from 1995 to 2023, examining communal media's impact on psychological concerns and highlighting methodological insights. Subsequently, the ''Results and Discussion of Real-world Survey'' section analyzes demographic data, usage patterns, and user perceptions from a survey focused on platforms

like TikTok. This section also discusses correlations among survey responses, providing insights into user behaviors and attitudes.

# METHODOLOGY

This section outlines the methodology used to analyze human communal media interaction and psychological impacts based on survey data collected from various sources.

## Considerations

This work employs a systematic approach to identify, select, and analyze studies exploring the human communal media interaction and its impact on mental concerns. The following steps were taken to ensure a rigorous and unbiased review:

1. Search strategy: comprehensive strategy targeted academic databases (PubMed, PsycINFO, Google Scholar) using keywords like social and communal media, mental and psychological concerns, and health. Also with specific platform names. Emphasis was placed on relevant datasets.
2. Inclusion and exclusion criteria: Inclusion criteria covered peer-reviewed, English-language studies focusing on the communal media-psychological concerns relationship, excluding non-relevant or poorly reported studies.
3. Data extraction: Data extracted independently on study design, demographics, communal media platforms, key findings, and statistical analyses.
4. Quality assessment: Studies underwent quality assessment based on design, sample size, statistical methods, and result clarity, with higher-quality studies given more weight.
5. Data synthesis and analysis: Synthesized data were thematically analyzed to identify patterns, trends, and contradictions, categorizing studies based on key themes and considering both positive and negative effects.
6. Risk of bias assessment: Thorough bias assessment included selective reporting, publication bias, and methodological limitations, with sensitivity analyses performed.
7. Search timeline: The search spanned 1995 to November 2023, ensuring the review incorporates the latest available studies.
8. Real world survey and ethical statements: This study also reveals the dynamics of social media communities through the implementation of a real-world survey which is discussed in a later section. All the respondents in the survey conducted in this research gave full consent to work with their data. No respondent's identity is disclosed. This study was approved by the Institutional Review Board of Jahangirnagar University, Dhaka, Bangladesh.

This systematic approach aims for a rigorous synthesis of the current knowledge on the communal media-psychological concerns relationship.

## Data collection of literary survey

Data for this study were obtained through a systematic review of literature and surveys conducted between 2000 and 2023. The survey data encompassed responses from diverse demographic groups, focusing on communal media usage patterns and their perceived psychological impacts.

## Literature review and categorization

A comprehensive literature review was conducted to identify relevant studies on human communal media interaction and psychological concerns. The studies were categorized into sixteen distinct categories, including emotional sentiment analysis, cultural dimensions, and effects during the COVID-19 pandemic.

## Qualitative analysis

Qualitative analysis of open-ended survey responses provided insights into nuanced perceptions and experiences of communal media usage and psychological well-being. Thematic analysis techniques were employed to identify recurring themes and qualitative patterns among respondents' narratives.

## Visualization techniques

Visualization techniques, including word clouds and keyword co-occurrence networks, were utilized to illustrate prevalent themes and relationships derived from the survey data and literature review. These visual representations aided in synthesizing complex data patterns and communicating findings effectively.

## Methodology and demographics of the real-world survey

To recognize the real-world scenarios, we conducted a comprehensive survey with a sample size of 140 respondents (only 86 respondents answered all the survey questions), aged between 15 and 26, representing the youth demographic in Bangladesh. The respondents were from various educational institutions nationwide, including schools, colleges, and universities. Among the respondents, 57% identified as female and 43% as male, with the majority falling within the age range of 21–24 years old. In the survey, there were a total of 23 questions where no questions were required (anyone can answer or ignore any questions if they wish). Written consent was obtained from all the respondents using Google Forms (Supplemental Information 3). We are grateful to them for their valuable information. This study was approved by the Institutional Review Board of Jahangirnagar University, Dhaka, Bangladesh, with IRB No. 479/IRB/JUH (September 14, 2021).

## Ethical considerations

Ethical considerations included obtaining informed consent from survey participants, ensuring confidentiality of responses, and adhering to ethical guidelines for research involving human subjects. By integrating quantitative and qualitative analyses of survey data and literature review findings, this study contributes to a deeper understanding of how communal media usage patterns influence psychological well-being across different demographic contexts.

## Correlation analysis of survey data

The survey data is represented as a matrix $X$, where each row $x_i$ corresponds to a respondent and each column represents a specific survey question or attribute. Let $x_{ij}$ denote the response of the $i$-th respondent to the $j$-th question. To normalize the data matrix $X$ to

ensure all variables are on the same scale, typically between 0 and 1 we follow:

$$x_{ij}^{\text{norm}} = \frac{x_{ij} - \min(X_j)}{\max(X_j) - \min(X_j)} \tag{1}$$

where $\min(X_j)$ and $\max(X_j)$ are the minimum and maximum values of column $j$ in matrix $X$.

After that, the missing values are handled with mean imputation as follows:

$$x_{ij} = \begin{cases} \text{mean}(X_j) & \text{if } x_{ij} \text{ is missing} \\ x_{ij} & \text{otherwise} \end{cases} \tag{2}$$

The mean vector $\bar{x}$ and covariance matrix $\Sigma$ of the normalized data $X$ can be obtained by:

$$\bar{x} = \frac{1}{n} \sum_{i=1}^{n} x_i \tag{3}$$

$$\Sigma = \frac{1}{n-1} \sum_{i=1}^{n} (x_i - \bar{x})(x_i - \bar{x})^T \tag{4}$$

where $n$ is the number of respondents.

Finally computing the correlation matrix $R$ to measure the linear relationships between variables is done by the following method:

$$R_{jk} = \frac{\sum_{i=1}^{n}(x_{ij} - \bar{x}_j)(x_{ik} - \bar{x}_k)}{\sqrt{\sum_{i=1}^{n}(x_{ij} - \bar{x}_j)^2}\sqrt{\sum_{i=1}^{n}(x_{ik} - \bar{x}_k)^2}}. \tag{5}$$

This evaluates correlations to identify significant associations between survey questions and demographic variables.

## RESULTS AND DISCUSSION OF LITERATURE SURVEY

The review provides a concise summary of key categories related to human-communal media interaction and psychological concerns. It encompasses a diverse range of topics, including the general impact of communal media, exploration of psychological disorders, examination of adolescent and young adult concerns, insights from longitudinal studies, and specific considerations such as the impact on sleep, social isolation, cultural dimensions, emotional sentiment analysis, technostress, and effects during the COVID-19 pandemic. This work also includes references to relevant studies, offering a comprehensive resource for those interested in the complex interplay between communal media use and psychological well-being. This work also unveils communal media dynamics by conducting a real-world survey. The survey reveals a diverse user landscape with differing views on safety, privacy, and the impact of the selected communal media platform on creativity and mental health. In this work, communal media means all types of online social networking sites and communication mediums.

## Categorical influence of human communal media interaction and mental concerns

The intricate relationship between communal media and psychological concerns is a focal point of recent research, categorized into distinct areas as illustrated in Fig. 1.

The summary of the categories on communal media and psychological concerns is shown in Table 1.

A brief explanation of the categorical studies is as follows:

### Psychological disorders for communal media

When we talk about psychological disorders, we're delving into the complexities of our minds and emotions. Researchers are using systematic reviews to uncover detailed information about how using communal media might affect our mental well-being. It's like shining a light on the various methods and discoveries related to the connection between communal media and our minds. Imagine these reviews as careful examinations, like detectives investigating a case. They help us see the bigger picture by gathering and analyzing all the evidence (*Wongkoblap, Vadillo & Curcin, 2017*). The goal is to understand the intricate details and to see how our use of communal media may influence our mental health. It's like trying to understand the relationship between two things—communal media and our well-being. But, as we explore this fascinating realm, there's another equally important aspect: ethics. Ethics here means doing this research responsibly and fairly (*Sadagheyani & Tatari, 2020*). It's like ensuring that detectives follow the rules and treat everyone fairly during an investigation. In the world of researching communal media and mental health, ethics remind us to be careful and considerate. Researchers, like *Wongkoblap, Vadillo & Curcin (2017)* and *Conway & OConnor (2016)* emphasize the significance of ethics. It's like they're telling us, "Let's not just find answers; let's find them in the right way." *Nicholas, Onie & Larsen (2020)* also highlight the ethical side, pointing out that it's not just about what we discover but how we go about discovering it. So, when we explore the realm of psychological concerns disorders, and communal media, it's not just about facts and figures. It's about being ethical detectives, carefully studying the details, and making sure our exploration is responsible and fair. It's like taking a journey into the mysteries of the mind, making sure we uncover the truth in the right way.

### Psychological concerns among adolescents for communal media

Let's look at how communal media affects the minds of teenagers. It's like exploring a landscape with both good and not-so-good things happening. There's a mix of positive and negative influences on adolescent psychological concerns. Think of it as trying to understand the complicated world of online interactions. *Oreilly et al. (2018)*, *Oreilly (2020)* and *Haddad et al. (2021)* analyzed the details to get a full picture of how communal media impacts the mental well-being of teenagers. It's a bit like navigating a path where some parts are bright and cheerful, while others might have challenges. By studying this landscape, we can see how communal media shapes the thoughts and feelings of adolescents. Adolescents' minds are very sensitive to becoming affected by communal media. So, when we talk about psychological concerns among adolescents and communal media, it's like

## Categories of Social Media Impact on Mental Health

**Figure 1** Categories of human communal media interaction and its impact on psychological concerns.

deciphering a code. How communal media affect teenagers' mental condition and how we may meditate on them are analyzed by *Boer et al. (2021)*.

### Young adults psychological conditions for communal media

Let's dive into how communal media affects the mental well-being of young adults. *Berryman, Ferguson & Negy (2018)* investigated the condition of poor young adults for the screen time they give to communal media. The results are not positive. The more they use communal media the more they waste their valuable time. The impact of communal media on young adults also has both positive and negative consequences (*Coyne et al., 2020*). The time spent by young adults in communal media is huge. These times have a huge impact on their daily life, mental conditions, social status, and career (*Strickland, 2014*). It's like solving a puzzle to understand the intricate relationship between using communal media and the outcomes on the mental health of young adults. Why is this exploration important? Well, it's about recognizing the unique challenges that young adults face. By understanding these challenges, we can create specific plans and strategies to help them navigate the digital landscape. It's like developing a road map tailored to the needs of young adults, ensuring they have the support and guidance necessary for a positive experience on communal media (*Cain, 2018*).

**Table 1  Summary of categories on human communal media interaction and psychological concerns.**

| No. | Category | Key Findings | References |
|---|---|---|---|
| 1. | General Impact | Transformation of communication, both benefits and risks explored | – |
| 2. | Psychological disorders | Exploration of factors in the complex relationship | (*Wongkoblap, Vadillo & Curcin, 2017*; *Sadagheyani & Tatari, 2020*; *Conway & OConnor, 2016*; *Nicholas, Onie & Larsen, 2020*) |
| 3. | Adolescent psychological concerns | Nuanced exploration of positive and negative aspects in online interactions | (*Oreilly et al., 2018*; *Oreilly, 2020*; *Haddad et al., 2021*; *Boer et al., 2021*) |
| 4. | Young Adults | Mixed consequences, associations between communal media engagement and psychological concerns outcomes | (*Berryman, Ferguson & Negy, 2018*; *Coyne et al., 2020*; *Strickland, 2014*; *Cain, 2018*; *Boer et al., 2021*; *Barthorpe et al., 2020*) |
| 5. | Students | Unique dynamics of students' interaction with communal media | (*Strickland, 2014*; *Wongkoblap, Vadillo & Curcin, 2017*; *Abi-Jaoude, Naylor & Pignatiello, 2020*) |
| 6. | Longitudinal Studies | Insights into temporal dynamics, associations over time | (*Coyne et al., 2020*; *Chou, Liang & Sareen, 2011*; *Scott & Woods, 2019*; *Bekalu, McCloud & Viswanath, 2019*) |
| 7. | Online Social Networking | Initial insights into dynamics between virtual interactions and mental well-being | (*Pantic, 2014*; *Brooks, Longstreet & Califf, 2017*; *Karim et al., 2020*; *Conway & OConnor, 2016*; *Nicholas, Onie & Larsen, 2020*) |
| 8. | Using Intensity | Examination of frequency and depth of engagement, effects on well-being | (*Liu et al., 2021*; *Boer et al., 2021*) |
| 9. | Impact on Sleep | Exploration of links between communal media use, sleep, and psychological concerns | (*Scott & Woods, 2019*; *Kelly et al., 2018*) |
| 10. | Social Isolation and Loneliness | Complex exploration of roles predicting loneliness and depressed mood | (*Boivin, Hymel & Bukowski, 1995*; *Chou, Liang & Sareen, 2011*; *Ayers et al., 2013*) |
| 11. | Cultural Impact | Examination of Social Mobile Media Culture and technology portrayal | (*Malathy M., 2018*) |
| 12. | Emotional Sentiment Analysis | Application of sentiment analysis techniques for psychological concerns safety | (*Benrouba & Boudour, 2023*; *Benrouba & Boudour, 2022*) |
| 13. | Technostress and Internet Addiction | Investigation of communal media-induced technostress and internet addiction | (*Brooks, Longstreet & Califf, 2017*; *Liu et al., 2021*) |
| 14. | Smartphones and psychological concerns | Scrutiny of smartphone usage patterns, communal media engagement, and psychological concerns outcomes among youth | (*Abi-Jaoude, Naylor & Pignatiello, 2020*; *McCrory, Best & Maddock, 2020*) |
| 15. | Effects during the COVID-19 Pandemic | Review of pandemic's impact on college psychological concerns, exploration of emotional response to COVID-19 information | (*Litwin & Levinsky, 2022*; *Haddad et al., 2021*; *Jones, Mougouei & Evans, 2021*) |
| 16. | Economic dimension | Analysis of mental health and economic dimensions | (*Braghieri, Levy & Makarin, 2022*; *Karim et al., 2020*) |

### Psychological concerns among students for communal media

The effect of communal media on students is mostly negative rather than positive. *Boer et al. (2021)* investigated the problems and suggested some solutions to these. Smartphones and communal media have a huge impact on the mental health and academic results of the students (*Boer et al., 2021*; *Deepa & Priya, 2020*). For students facing the ups and downs of school life, it's crucial to grasp how they connect with communal media. Understanding

this unique relationship is essential. We need to dig into how online platforms influence students' experiences and, in turn, impact their mental well-being (*Strickland, 2014*; *Wongkoblap, Vadillo & Curcin, 2017*; *Abi-Jaoude, Naylor & Pignatiello, 2020*). Think of it like students embarking on a journey through the digital world. We're trying to figure out how their encounters on communal media platforms shape the way they feel and think. It's a bit like uncovering the secrets of how online activities influence the daily lives of students. This exploration helps us understand the connection between students and communal media, especially in the context of their mental well-being.

### Longitudinal studies for communal media and psychological concerns

*Coyne et al. (2020)* investigated the impact of communal media on mental health for eight years and showed their results. They found long-time use of communal media does not have a good impact on depression and anxiety. They did their study on hundreds of persons including teenagers and young adults for eight years long. The study of *Chou, Liang & Sareen (2011)* is very vast among multiple aspects. They analyzed mental conditions, isolation, and alcohol impacts among Americans along with communal media impact. *Scott & Woods (2019)* investigated the impact of communal media on sleeping time and mental health. *Bekalu, McCloud & Viswanath (2019)* found pros and cons for mental health and communal media interaction. They also showed the emotional impact of social interaction on the internet. Embarking on longitudinal studies has proven instrumental in unraveling the temporal dynamics of communal media's impact on psychological concerns. These studies offer valuable insights into how the effects evolve over an extended period, contributing to a more comprehensive understanding.

### Online social networking and psychological concerns

Social networking sites and their impact on mental health are investigated by *Pantic (2014)*. Addiction to the internet and communal media has a bitter effect on our mental health. *Brooks, Longstreet & Califf (2017)* analyzed these effects in terms of internet addiction and technostress with the perspective of distraction and conflict. *Karim et al. (2020)* investigated fifty publications related to communal media and psychological health and showed some pros and cons. *Conway & OConnor (2016)* have gone through some big data analysis and ethical concerns along with online networking and mental concerns. Privacy concerns and ethics among social networking are responsible for the mental health of the users (*Nicholas, Onie & Larsen, 2020*). The exploration of online social networking unveils the initial insights into the intricate dynamics between virtual interactions and mental well-being. Delving into communal media-induced technostress and its psychological consequences emphasizes the need for ethical considerations in this evolving digital landscape.

### Using intensity of communal media and domestic violence

The intense use of communal media may have such impact on mental health that can lead a person to domestic violence (*Liu et al., 2021*). *Liu et al. (2021)* collected data from more than two hundred victims and found a relation between domestic violence and communal media. Examining the intensity of communal media use is crucial for understanding its potential impact on psychological concerns. This *Boer et al. (2021)* study investigated the

effects on psychological well-being for the frequency, depth of engagement, and intensity of communal media.

### Impact on sleep for communal media

Using communal media for a long time and in unusual times may affect our sleeping time broadly (*Scott & Woods, 2019*). Online interaction, time spent and mental involvement can cause low sleep or irregular sleep patterns in people. The relationship between communal media use, sleep, and psychological concerns unfolds as researchers explore the potential disruptions to sleep caused by online engagement. Understanding these effects is vital for developing informed guidelines and interventions (*Kelly et al., 2018*). *Twenge, Krizan & Hisler (2017)* found that using more communal media results in reduced sleep. The data set of *Perez-Lloret & Perez Chada (2022)* showed how the sleep pattern is affected by communal media use.

### Social isolation and loneliness for communal media

Though people use communal media for making friends and increasing social networking, overuse of communal media can make them socially isolated and lonely. Day by day a person addicted to communal media may be isolated from the real world and become a part of the virtual world. *Boivin, Hymel & Bukowski (1995)* and *Chou, Liang & Sareen (2011)* investigated how communal media is responsible for loneliness and depression. Early insights into predicting loneliness and depressed mood in childhood highlight the importance of considering social factors in psychological concerns outcomes. *Ayers et al. (2013)* dig into the deep of the topic and show the seasonal impact of communal media among the survey attendees. Delving into the relationship between communal media and its impact on social isolation and loneliness requires a multifaceted approach.

### Cultural impact for communal media

The cultural impact of communal media on psychological concerns is a distinctive avenue of exploration. *Malathy (2018)* and *LaRose et al. (2014)* portrayed how communal media made both positive and negative impacts on cultural contexts.

### Emotional sentiment analysis among communal media

As people are using communal media very intensely and giving lots of time in it they become very emotional and serious about it. *Benrouba & Boudour (2023)* analyzed the emotional aspects of human communal media interactions. The emotional analysis for both positive and negative impacts on mental health is shown by *Benrouba & Boudour (2022)*. Sometimes depressed people may also get some positive help from their communal media friends which may have some good impact on their mental health. communal media content for psychological concerns provides a unique view. Applying sentiment analysis techniques unveils patterns and indicators related to psychological concerns.

### Technostress and internet addiction for communal media

These studies (*Brooks, Longstreet & Califf, 2017*; *Liu et al., 2021*) underscore the potential negative consequences of over-communal media use. Using the high rate of communal media can make a person addicted to the internet. Internet addiction may cause

technostress. The above-mentioned studies showed how technostress is a result of communal media use and is harmful to human beings. Internet addiction may affect mental health which may make a person violent domestically (*Liu et al., 2021*). An internet-addicted person may lose control of his or her behaviors.

### Smartphones and psychological concerns in terms of communal media

Most of the people use communal media with smartphones. The long-time interaction with smartphone screens and communal media may cause harm to our eyesight and mental conditions (*Abi-Jaoude, Naylor & Pignatiello, 2020*). Highly visual communal media can create lots of impact on our mental conditions. A comprehensive exploration of the impact of smartphones and communal media use on youth psychological concerns is shown in this study (*McCrory, Best & Maddock, 2020*). Scrutinizing smartphone usage patterns, communal media engagement, and psychological concerns show the risks and benefits associated with modern technology.

### Effects during the pandemic for communal media

How communal media made an impact on people's mental health after the COVID-19 pandemic is illustrated by the *Litwin & Levinsky (2022)*. *Haddad et al. (2021)* showed a transnational analysis of communal media's impact on mental health at the time of the pandemic. How mental health and people's lives are affected by the communal media and news media are illustrated by *Jones, Mougouei & Evans (2021)*. At the time of the pandemic people could not go out of their houses and the communal media and news media became their only way of entertainment and information source. As a result, the communal media's positive impact made people's lives better and the negative impacts made harm to the mental health of the people during the pandemic time.

### Economic dimension for communal media and mental health

In today's world, almost every people are connected with communal media. So, news spread by communal media has great economic impacts. Economic changes in life and business made my communal media may harm mental conditions too. *Braghieri, Levy & Makarin (2022)* illustrated the multifaceted economic implications of communal media on psychological concerns. In their comprehensive analysis, they emphasize the necessity of understanding the complex interaction between communal media use and psychological concerns within the broader framework of economic dynamics. *Karim et al. (2020)* contribute to this understanding through a systematic study, shedding light on the relationships between communal media, psychological concerns, and various economic factors. Their synthesized overview provided valuable insights into how economic dimensions intersect with the communal media and psychological well-being.

## Research works on communal media and psychological concerns (1995–2023)

From the record of Google Scholar, the number of research works on human communal media interaction and psychological concerns from the year 1995 to 2023 are illustrated in Table 2.

**Table 2 Research works on the relationship between human communal media interaction and psychological concerns (1995–2023).**

| Time interval | Start year | End year | Research works |
|---|---|---|---|
| 1995–1999 | 1995 | 1999 | 272 |
| 2000–2004 | 2000 | 2004 | 459 |
| 2005–2009 | 2005 | 2009 | 1130 |
| 2010–2014 | 2010 | 2014 | 3530 |
| 2015–2019 | 2015 | 2019 | 3190 |
| 2020–2023 | 2020 | 2023 | 5200 |

Table 2 offers a chronological overview of research works on the relationship between communal media and psychological concerns spanning from 1995 to 2023. The data is categorized into five-year intervals to provide a comprehensive perspective on the evolution of scholarly interest in this domain. Between 1995 and 1999, the number of research works stands at 272, indicating a nascent exploration of the topic. Subsequent intervals reveal a steady increase, with significant growth observed in 2000–2004, 2005–2009, and 2010–2014, reaching 3,530 research works. The subsequent 2015–2019 period maintains high scholarly engagement with 3,190 works, showcasing sustained interest. Notably, the most recent interval from 2020 to 2023 exhibits a substantial surge, reaching 5,200 research works, underscoring the heightened attention and urgency surrounding the intersection of communal media and psychological concerns in recent years. This escalating trend may reflect the growing recognition of the profound impact of digital technologies on individuals' psychological well-being and underscores the critical importance of ongoing research in this dynamic field. The table serves as a valuable reference point for understanding the evolving landscape of research on communal media and psychological concerns, emphasizing the field's significance and the need for continued exploration in the years to come. The word cloud found in the examined research works is shown in Fig. 2. The most used 1,300 words were selected from the examined papers and then Python 3 visualization was used to illustrate this word cloud.

The keyword co-occurrence network (Fig. 3) visually maps the connections between key terms, offering a focused representation of associations in the field among the examined research works. This network map serves as a valuable tool for researchers, providing insights into prevalent themes and relationships in mental health literature, and guiding targeted explorations and further investigations. The top 50 most used keywords among the examined research work related to mental health and communal media are chosen to draw this network with Python 3 visualization.

After this, we consider sixteen categories for human communal media interaction and psychological concerns. The categories are explained before in Table 1. The category-wise research works found in Google Scholar are shown in Table 3.

The 2000–2023 analysis of communal media and psychological concerns uncovers key insights. Exploration into the psychological impact of excessive media use emphasizes the need for balanced technology consumption, supported by data indicating a significant rise from 2000 to 2019. Categories focused on adolescents and students reveal dynamic patterns

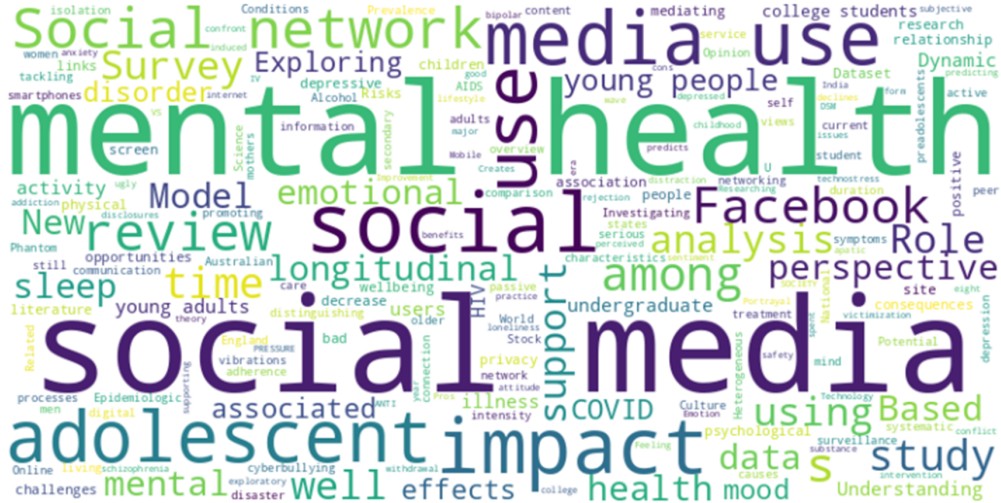

**Figure 2** Word cloud among the examined research titles of human social media interaction and psychological health (size of each word corresponds to its frequency).

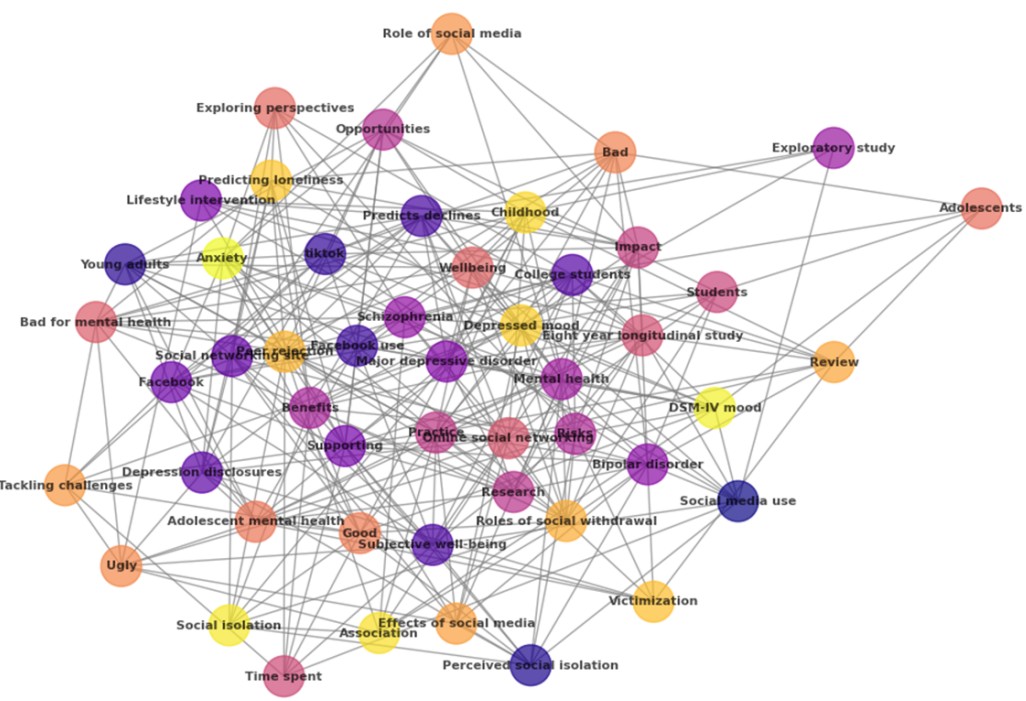

**Figure 3** Keyword co-occurrence network: exploring relationships among top 50 keywords in mental health and communal media research among the examined research works.

**Table 3  Category wise research works found in google scholar in the years (2000–2023) (see Table 1 for category explanations).**

| Categories | 2000–2004 | 2005–2009 | 2010–2014 | 2015–2019 | 2020–2023 |
|---|---|---|---|---|---|
| 1 | 3689 | 7220 | 14310 | 17220 | 9190 |
| 2 | 550 | 220 | 194 | 845 | 303 |
| 3 | 138 | 167 | 182 | 188 | 177 |
| 4 | 161 | 164 | 178 | 187 | 322 |
| 5 | 163 | 178 | 170 | 165 | 342 |
| 6 | 119 | 177 | 180 | 184 | 312 |
| 7 | 95 | 125 | 164 | 181 | 297 |
| 8 | 94 | 109 | 158 | 179 | 373 |
| 9 | 98 | 114 | 134 | 173 | 337 |
| 10 | 99 | 93 | 113 | 177 | 297 |
| 11 | 152 | 194 | 215 | 262 | 386 |
| 12 | 159 | 129 | 158 | 169 | 336 |
| 13 | 3 | 2 | 5 | 171 | 397 |
| 14 | 5 | 51 | 93 | 180 | 205 |
| 15 | 0 | 0 | 0 | 430 | 1070 |
| 16 | 167 | 179 | 178 | 180 | 537 |

**Table 4  Percentages of categorical impact from 2000 to 2023.**

| Year | Cat2 | Cat3 | Cat4 | Cat5 | Cat6 | Cat7 | Cat8 | Cat9 | Cat10 | Cat11 | Cat12 | Cat13 | Cat14 | Cat15 | Cat16 |
|---|---|---|---|---|---|---|---|---|---|---|---|---|---|---|---|
| 2000-2004 | 7.64% | 3.89% | 5.67% | 5.76% | 4.21% | 3.36% | 3.35% | 3.49% | 3.50% | 5.37% | 5.58% | 0.08% | 0.13% | 5.90% | 1.17% |
| 2005-2009 | 3.33% | 4.69% | 5.53% | 6.24% | 5.52% | 4.65% | 3.97% | 4.28% | 4.13% | 8.17% | 5.65% | 0.03% | 0.26% | 6.38% | 1.48% |
| 2010-2014 | 2.92% | 5.13% | 5.33% | 5.84% | 5.68% | 6.11% | 5.87% | 4.58% | 4.89% | 6.93% | 5.47% | 0.13% | 0.41% | 7.15% | 1.91% |
| 2015-2019 | 12.57% | 5.51% | 5.59% | 5.62% | 5.64% | 6.48% | 6.45% | 5.69% | 5.39% | 7.35% | 5.11% | 11.77% | 10.14% | 5.36% | 2.55% |
| 2020-2023 | 4.49% | 5.21% | 8.89% | 5.83% | 5.55% | 6.32% | 6.19% | 5.38% | 6.51% | 6.97% | 4.92% | 14.91% | 12.50% | 10.11% | 3.18% |

in research attention over the years. Longitudinal studies offer valuable temporal insights, though variations in attention across periods are apparent. Investigations into media engagement intensity, sleep impact, and feelings of isolation highlight nuanced trends. Cultural and economic dimensions, reflected in varying research focus, underscore societal implications. The surge in emotional sentiment analysis and pandemic-related studies reflects adaptive research, with evident recent upward trends. Attention to smartphones and emerging trends signals awareness of evolving technology patterns, especially among the younger demographic.

Table 4 illustrates the evolving percentages of categorical impact from 2000 to 2023. The values in each cell are computed using the following formula:

$$\text{Percentage} = \left( \frac{\text{Number of research works in the category for a given year}}{\text{Total number of research works for that year}} \right) \times 100. \qquad (6)$$

For instance, the percentage for Cat2 in the year 2000 is calculated as follows:

$$\text{Cat2 percentage in 2000} = \left( \frac{\text{Number of Cat2 research works in 2000}}{\text{Total number of research works in 2000}} \right) \times 100. \qquad (7)$$

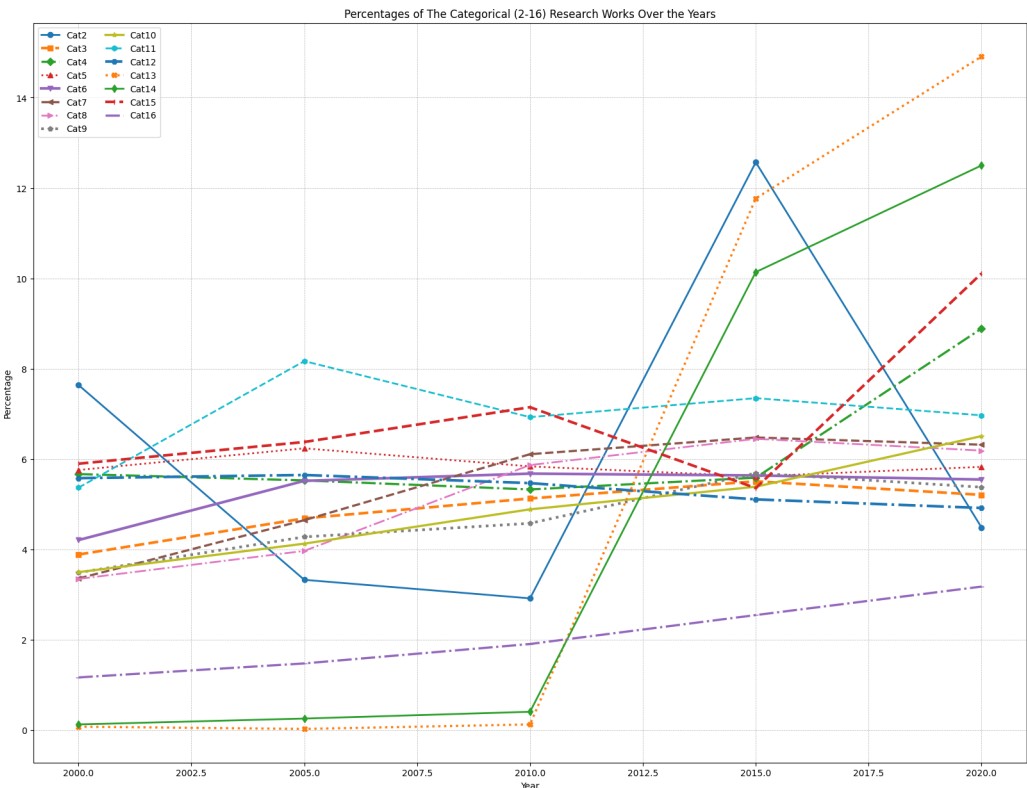

**Figure 4** **Percentage wise categorical (cat2 to cat16) research works over the years (2000–2023) in Google Scholar.**

Figure 4 shows percentage-wise categorical (cat2 to cat16) research works over the years (2000–2023) found in Google Scholar.

These percentages provide insights into the relative emphasis on each category over time. The trends indicate notable shifts in research focus, such as the increasing attention to categories like Emotional Sentiment Analysis (Cat12) and the Economic Dimension (Cat16) in recent years. The calculated percentages serve as a valuable quantitative representation of the distribution of research efforts across various categories, offering a nuanced perspective on the evolving landscape of communal media and psychological concerns research.

# RESULTS AND DISCUSSION OF REAL-WORLD SURVEY

## Data and questions

The survey questions and results are shown in Table 5. Here, Q = Question. There were a total of 23 questions. The percentages are computed based on the analysis of participation in each question and the corresponding responses.

## Survey outcomes

The survey results provide a comprehensive understanding of user perspectives on the communal media TikTok. Let's delve into key themes emerging from the responses:

**Table 5  Summary of the survey responses.**

| Question | Answer | Percentage | Respondents |
|---|---|---|---|
| | a. Female | 55.80% | 48 |
| Q1: Gender | b. Male | 43.00% | 37 |
| | c. Preferred not to say | 1.20% | 1 |
| | a. Below 16 years old | 2.30% | 2 |
| Q2: Age Range | b. 16-20 years old | 29.10% | 25 |
| | c. 21-24 years old | 64.00% | 55 |
| | d. 25 and above | 4.70% | 4 |
| Q3: TikTok App Download | a. No | 51.20% | 44 |
| | b. Yes | 48.80% | 42 |
| | a. Regularly use for both watching and making videos | 10.50% | 9 |
| Q4: Frequency of TikTok Use | b. Use regularly but only to watch videos | 16.30% | 14 |
| | c. Barely use the app | 34.90% | 30 |
| | d. Never used TikTok | 38.40% | 33 |
| | a. Very much | 14.00% | 12 |
| Q5: Enjoyment of TikTok | b. A little | 23.30% | 20 |
| | c. Neutral | 32.60% | 28 |
| | d. Dislikes it | 30.20% | 26 |
| | a. Making videos on TikTok lets me be creative | 8.10% | 7 |
| Q6: Why Use TikTok | b. TikTok helps me stay occupied during my leisure time | 29.10% | 25 |
| | c. Get to learn exciting hacks and facts from TikTok | 34.00% | 30 |
| | d. Discover trending music from TikTok | 29.10% | 25 |
| | a. Less than an hour | 70.90% | 61 |
| Q7: Time Spent on TikTok | b. 1-2 h | 23.30% | 20 |
| | c. 3-5 h | 3.50% | 3 |
| | d. More than 5 h | 2.30% | 2 |
| | a. Lip syncing videos | 8.10% | 7 |
| Q8: Type of TikTok Videos | b. Dance videos | 10.50% | 9 |
| | c. Food reviewing videos | 5.80% | 5 |
| | d. Comedy | 5.80% | 5 |
| | a. Features and effects make it easy to create high-quality videos | 31.40% | 27 |
| Q9: Why Prefer TikTok | b. Free trending platform catering content to preferences | 40.70% | 35 |
| | c. Widely used making it easier to make friends or find/share videos | 27.90% | 24 |
| | a. Works as a distraction from external worries | 23.30% | 20 |
| Q10: Positive Impacts of TikTok | b. Inspires people to try new creative ideas | 38.40% | 33 |
| | c. Helps create friends | 7.00% | 6 |
| | d. Excellent source of entertainment | 50.00% | 43 |
| | a. Yes | 11.60% | 10 |
| Q11: Safety of TikTok | b. No | 27.90% | 24 |
| | c. Maybe | 60.50% | 52 |

**Table 5** (*continued*)

| Question | Answer | Percentage | Respondents |
|---|---|---|---|
| Q12: Negative Impacts of TikTok | a. Invasion of privacy | 23.30% | 20 |
| | b. Harmful to underaged users | 55.80% | 48 |
| | c. Creates social nuisance due to addiction | 45.30% | 39 |
| | d. Platform where people can be bullied or harassed | 38.40% | 33 |
| Q13: Impact on Depression | a. Helps people unwind and feel entertained | 17.40% | 15 |
| | b. Not sure about its contribution to depression | 59.30% | 51 |
| | c. Creates more social comparison affecting mental health | 23.30% | 20 |
| Q14: Influence on Physical Appearance | a. Yes | 10.50% | 9 |
| | b. Sometimes | 19.80% | 17 |
| | c. Never | 69.80% | 60 |
| Q15: Teenagers' Addiction Impact | a. Strongly Disagree | 15.10% | 13 |
| | b. Disagree | 33.70% | 29 |
| | c. Agree | 5.80% | 5 |
| | d. Strongly Agree | 45.30% | 39 |
| Q16: Disturbing Videos Encounter | a. Yes | 3.50% | 3 |
| | b. No | 5.80% | 5 |
| Q17: Bullying/Scam Experience | a. Yes | 19.80% | 17 |
| | b. No | 43.00% | 37 |
| Q18: Overall Impression of TikTok | a. Productive | 34.90% | 30 |
| | b. Waste of time | 40.70% | 35 |
| | c. Harmful | 24.40% | 21 |
| | d. No comments | 1.20% | 1 |
| Q19: Bad Impact on Underaged Users | a. Strongly Disagree | 32.60% | 28 |
| | b. Disagree | 16.30% | 14 |
| | c. Agree | 75.60% | 65 |
| | d. Strongly Agree | 17.40% | 15 |
| Q20: Satisfaction with TikTok Policies | a. Yes | 66.30% | 57 |
| | b. No | 62.80% | 54 |
| Q21: Harmful App Features | a. Anonymity | 19.80% | 17 |
| | b. Filters | 41.90% | 36 |
| | c. Sharing personal data to third-party apps | 12.80% | 11 |
| | d. Free to download | 34.90% | 30 |
| Q22: Safer Platform Suggestions | a. Verification of Age and Identity during account opening | 32.60% | 28 |
| | b. Strict actions against users who violate rules | 16.30% | 14 |
| | c. Turning it into a paid app | 75.60% | 65 |
| | d. Allowing the option of controlling who can use your data | 17.40% | 15 |
| Q23: TikTok Ban Controversy | a. Support the ban of TikTok | 34.90% | 30 |
| | b. Believe the app could make amends | 40.70% | 35 |
| | c. Feel it is unnecessary | 24.40% | 21 |

1. Demographics: The survey reveals that the majority of users fall within the 16–24 age range, with 55.80% identifying as female and 43.00% as male. The app seems particularly popular among young adults, with 64.00% of respondents aged 21–24.

2. App usage and preferences: Approximately half of the respondents have not downloaded the app (51.20%), while 48.80% have. Among the users, 34.90% barely use the app, and 38.40% have never used it. Reasons for using the app include leisure (29.10%), creativity (8.10%), and content discovery (34.00%).

3. Positive and negative impacts: Users report positive impacts such as distraction from external worries (23.30%) and inspiration for creative ideas (38.40%). However, concerns arise regarding invasion of privacy (23.30%) and harm to underaged users (55.80%).

4. Safety and privacy concerns: Regarding safety, 60.50% of users are uncertain, while 27.90% express no safety concerns. Privacy concerns include anonymity (19.80%) and sharing personal data with third-party apps (12.80%). A significant 41.90% are wary of filters.

5. Societal and ethical aspects: Opinions on the app's impact on youth are divided, with 75.60% agreeing that it has a significant impact. A substantial portion (40.70%) believes the app is a waste of time, while 34.90% view it as productive. Suggestions for a safer platform include age verification (32.60%) and turning it into a paid app (75.60%). The results from Q4 to Q23 are illustrated in Fig. 5.

Finally, the survey results portray a diverse landscape of user experiences and opinions. Developers and policymakers should consider the varied perspectives on safety, privacy, and the app's impact on creativity and mental health. Ongoing discussions and adaptations are crucial to align the platform with evolving user expectations and concerns.

## Results of correlation analysis

The correlation matrix (Fig. 6) visualizes the relationships between survey questions q1 to q23. The values of q1 to q23 are mentioned in Table 5. It helps identify patterns of positive and negative associations among variables, highlighting key dependencies that influence human communal media interaction and psychological concern. Here we can see, that q2 shows strong positive correlations with q5. q1 and q4 exhibit negative correlations. Variables q16, q17, and q20 show a strong negative correlation with q3, q16, and q17. The breakdown and insights of the correlations between each pair of survey questions (q1 to q23) are as follows:

- Positive correlations (values close to 1): Questions like q2, q5, q8, q10, q12, q14, q16, q17, q20, and q22 show strong positive correlations with themselves and similar questions. For example, q2, q5, and q8 are highly correlated (1.000) because they likely measure similar aspects or have similar response patterns among respondents.
- Negative correlations (values close to -1): Negative correlations are observed, but less prominently. For example, q1 and q4 show a strong negative correlation of −0.870.
- Zero or weak correlations (values close to 0): Questions like q9, q11, q13, q15, q18, q19, q21, and q23 show weaker correlations with other questions or even no correlation in some cases (values like 0 or very close to it).

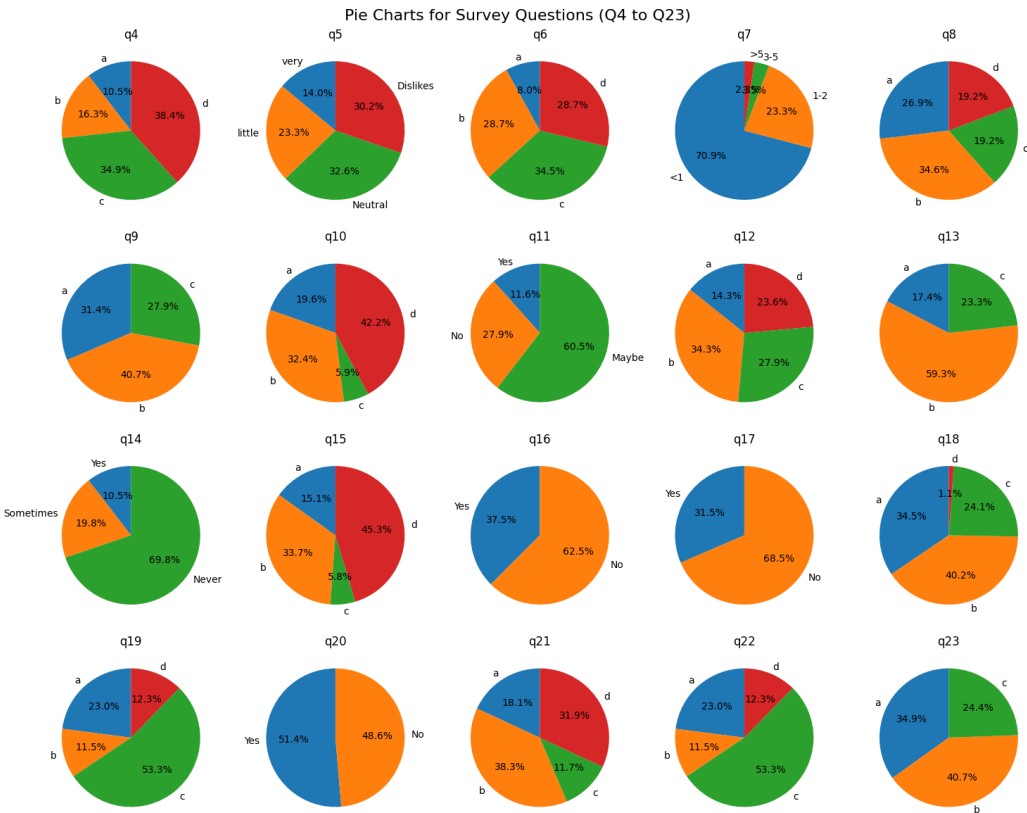

**Figure 5  Summary results from Q4 to Q23 of the survey.** The elaboration of Q4 to Q23 is presented in Table 5.

- Question clusters: Based on the correlations, we can identify clusters of questions that tend to correlate strongly together. For instance, q2, q5, and q8 form a cluster with high positive correlations, indicating they might address similar themes or aspects of the survey.
- Contrasting questions: Questions like q1 and q4, with their strong negative correlation, might represent opposite ends of a spectrum or measure contrasting aspects among respondents.
- Independence: Questions with correlations close to zero (*e.g.*, q9, q11, q13, q15, q18, q19, q21, q23) suggest that they may be independent of each other in terms of respondent perceptions or behaviors.
- Dimension reduction: Identifying highly correlated questions can help in reducing the number of variables in further analyses without losing significant information.
- Identifying confounding factors: Negative correlations can highlight potential confounding factors or opposite effects within the survey data.

This matrix serves as a powerful tool for understanding relationships between survey questions, potentially guiding how to group or analyze them in subsequent statistical or clustering analyses.

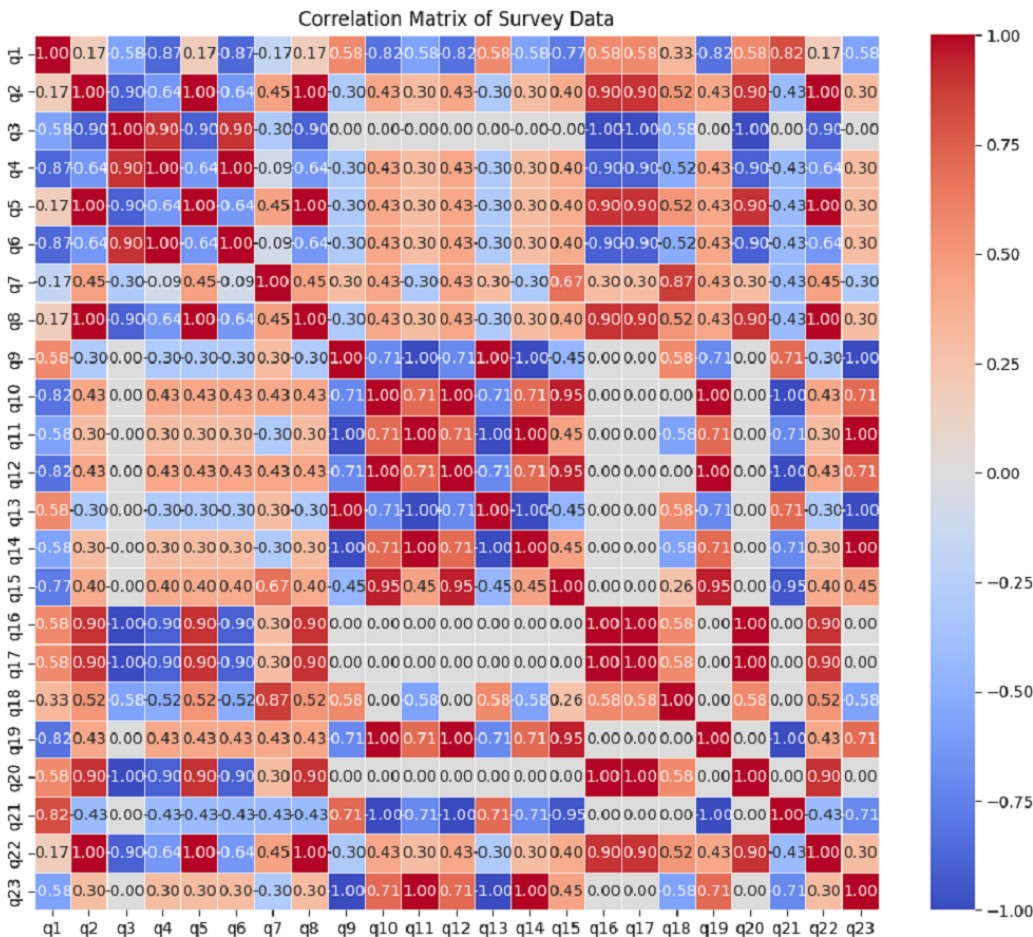

**Figure 6** Correlation matrix of survey data (here, the values of q1 to q23 are mentioned in Table 5).

## CONCLUSIONS

In conclusion, the comprehensive literature review has provided a nuanced understanding of the evolving relationship between communal media and psychological concerns within human–computer interaction. Employing a methodologically rigorous human–computer interaction approach, we systematically categorized research works from 1995 to 2023, revealing a substantial increase. This upward trend underscores the growing recognition of communal media's profound impact on psychological concerns, reflecting the dynamic nature of the human–computer interaction landscape. Our analysis delved into diverse human–computer interaction aspects, including effects on demographic groups, emotional sentiment analysis, online social networking, and communal media use intensity, illuminating the multifaceted dimensions of this relationship. This work also unveils communal media dynamics by conducting a real-world survey. The survey reveals a diverse user landscape with differing views on safety, privacy, and the impact of the selected communal media platform on creativity and mental health. The survey results provide a nuanced understanding of TikTok users' perspectives, revealing the platform's positive

and negative impacts. The data highlights a demographic predominance of young adults aged 16–24, with a significant gender representation favoring females. Usage patterns show a division between users and non-users, with creativity, leisure, and content discovery being primary motivators for engagement. Positive impacts such as creative inspiration and distraction from worries contrast sharply with concerns over privacy invasion, harm to underaged users, and addiction. Safety and privacy remain contentious issues, with a substantial portion of users expressing uncertainty or concern. The correlation analysis underscores the complex interplay between various user experiences and perceptions, identifying clusters of related survey questions that reflect common themes. Overall, the findings underscore the need for developers and policymakers to address diverse user concerns, balancing the creative and entertainment value of TikTok with robust safety, privacy, and ethical considerations.

Future research should focus on enhancing privacy and safety measures, such as developing more stringent age verification processes and advanced privacy settings. Educational campaigns can inform users about the potential risks and safe practices associated with TikTok, particularly targeting younger demographics. In-depth studies on the long-term psychological impacts of TikTok usage, such as addiction and mental health, are essential. Platform modifications, including the feasibility of transitioning to a paid app model and implementing stricter content moderation policies, should be explored. Longitudinal studies can track changes in user behavior over time, while cross-platform comparisons with other social media platforms can inform best practices for social media design and regulation. Addressing these areas will help create a safer, more engaging, and ethically sound social media environment, ensuring that platforms like TikTok continue to offer value to users while mitigating potential negative impacts. Looking forward, the future of research in this domain presents promising avenues. Identified gaps, particularly in exploring specific demographic groups and developing predictive models, offer opportunities for deeper investigations. Advanced statistical techniques and machine learning algorithms can refine predictions, providing a comprehensive understanding of temporal dynamics. This work significantly contributes to the expanding body of knowledge, providing a robust foundation for future research to inform policies, interventions, and strategies promoting positive psychological outcomes in the digital age.

### Funding
The authors received no funding for this work.

### Competing Interests
Deep Ajabani is employed by Source InfoTech Inc. Loganville, Georgia, United States.

## Author Contributions

- Tajim Md. Niamat Ullah Akhund conceived and designed the experiments, performed the experiments, performed the computation work, prepared figures and/or tables, authored or reviewed drafts of the article, and approved the final draft.
- Deep Ajabani conceived and designed the experiments, performed the experiments, analyzed the data, performed the computation work, prepared figures and/or tables, authored or reviewed drafts of the article, and approved the final draft.
- Zaffar Ahmed Shaikh conceived and designed the experiments, analyzed the data, performed the computation work, prepared figures and/or tables, authored or reviewed drafts of the article, and approved the final draft.
- Ali Elrashidi conceived and designed the experiments, performed the experiments, analyzed the data, performed the computation work, prepared figures and/or tables, authored or reviewed drafts of the article, and approved the final draft.
- Waleed A. Nureldeen conceived and designed the experiments, performed the experiments, performed the computation work, prepared figures and/or tables, authored or reviewed drafts of the article, and approved the final draft.
- Muhammad Ishaq Bhatti conceived and designed the experiments, analyzed the data, performed the computation work, authored or reviewed drafts of the article, and approved the final draft.
- M. Mesbahuddin Sarker conceived and designed the experiments, performed the experiments, performed the computation work, authored or reviewed drafts of the article, and approved the final draft.

## Ethics

The following information was supplied relating to ethical approvals (i.e., approving body and any reference numbers):

The Institutional Review Board of Jahangirnagar University granted ethical approval to carry out the study within its facilities (Reference No. 479/IRB/JUH; Dated: September 14, 2021).

## Data Availability

The data and code are available in Supplementary Files 1 and 2.

## Supplemental Information

Supplemental information for this article can be found online at http://dx.doi.org/10.7717/peerj-cs.2398#supplemental-information.

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
