# Peer review of "A comprehensive exploration of human communal media interaction and its evolving impact on psychological health across demographics and time"

_PeerJ Computer Science, doi:10.7717/peerj-cs.2398_

## Round 0.1 · original submission · Major Revisions

The reviewers have provided detailed comments and suggestions to improve the quality and clarity of your manuscript. We kindly ask you to address each of the reviewers' comments thoroughly in your revised manuscript.

Here, I summarize the main points.

Abstract and Paper Flow: Improve the flow and structure of the paper and ensure the abstract clearly outlines the scope of the work. The abstract needs to be revised to include three parts, and the method section must be clarified.

Strengthen the rationale for the study.

Method: Provide methodological details. The method used in lines 456-465 is inappropriate; justification and details about the predicted and validated data set are needed. Emphasize statistical analysis and mathematical modelling.

Validity of Findings: Explain the predicted and validated data sets and the percentage error encountered. Ensure transparency and replicability of the research.

Improve descriptions and explain images according to the reviewers’ comments.

Reviewer 1 ·

Basic reporting

The abstract must contain 3 parts, revise the method section used

Experimental design

It is hoped that the existing image can be clarified

Validity of the findings

Please explain the calculation formula from the existing data in the Future Aspects of Communal Media and Psychological Concerns: A Mathematical Per 455 perspective

Additional comments

The title is appropriate and relevant to the content of the research paper

Annotated reviews are not available for download in order to protect the identity of reviewers who chose to remain anonymous.

·

Basic reporting

I praise the authors for their extensive data set, compiled over many years of detailed field work and real survey work. In addition, the manuscript is clearly written in professional, unambiguous language. If there is a weakness, it is in the mathematical analysis (as I have noted above) which should be improved upon before Acceptance.

1) M.A Moreno et al (J of Depress Anxiety, 2011, 28(6) pp. 447–455) have shown statistical analysis using STATA. You use in Lines 456 – 465 is not the most appropriate for this situation. Please explain why you used this method. Also please explain where is your predicted and validated data set.

2) Your mathematical perspective needs more detail. I suggest that you improve the description at lines 32 – 34, 466 – 468 and 482 – 488 to provide more justification for your study.

Experimental design

3) I thank you for providing the raw data, however your supplemental files need more descriptive metadata identifier to be useful to future readers. Although your results are compelling, the data analysis should be improved in the following ways:
(a) In Figure 3 and data table 3 are controversial because you are saying data at 2020-2023 is 5200 research works but in figure 3 it is in reverse trend. Please correct it.
(b) In Line 420 and for Figure-4, there are 1300 words were selected from the examined papers, is there any specific reason to select these many words on what basis the frequency of the word is set, it has to be explained. Otherwise instead of 1300 words may be most important words (say 300 words) could be selected to show words frequency factor.
(c) Table 4 and Figure 6 reveals the same meaning so anyone can be kept, most probably Table 4 is meaningful.
(d) Table 5 and Figure 7, how did you arrive % of categorical impact, which data point did you used for calculation please explain. The total percentage contribution /impact of all categories will be 100% or the summation should be up to 100%. Or otherwise either in row or column wise if you add it should come 100%.
(e) Effect of each category to be connected with human communal media.
(f) From the Figure 6 and Figure 7; it would be more appropriate to do curve fitting and have the regression equations and then use that equations for developing a model and then predict and then validate.

Validity of the findings

Please explain where your predicted and validated data sets are.
What is the percentage error you came across?

Additional comments

The authors have made more effort on data collection, compilation, and field survey but the real statistical analysis and mathematical modeling are missing!

Reviewer 3 ·

Basic reporting

Reporting and writing style is consistent and adheres to good quality academic English. However, I would suggest to improve the flow of the paper as there are many headings, many parts to the work and it is truly hard as a reader to follow the argument. Moreover, abstract is not fully discussing what the paper is about. It is instrumental to clearly write the scope of the work in the abstract and then structure paper so it delivers a more engaging and logical flow. To start with I thought this work is a systematic lit.review but then I would expect methodological details on that - obviously after seeing survey results, I realised there is more to paper. Clearly, you need to properly communicate everything. IF you are reporting systematic review results, please provide all details to instruments for quality assessment, keywords identification and all key methodological steps. Descriptive primary dataset is interesting but to be absolutely frank I do not see what value such basic analysis has. Justification for this research needs strengthening - to be absolutely frank I do not see a strong rationale for this study.

Experimental design

Overall, methodologically this work is truly weak. No details into methodological septs for both, systematic review and survey. Overall, I was confused by purpose of this study and what is aims to achieve.

Validity of the findings

I would not consider systematic review results being truly rigorous as at this stage details are not reported - so transparency and replicability of the research are weak. Results of survey are largely descriptive and at times I am not sure what the findings are adding as new knowledge.

Additional comments

Overall, I ding this paper hard to understand and of course much of details are missing including a clearer and stronger rationale for this work.

---

## Round 0.2 · accepted · Accept

The paper has been revised to address and incorporate the reviewers' comments. The new version can be accepted.

·

Basic reporting

it is ok

Experimental design

it is ok

Validity of the findings

fine

Additional comments

nothing